# Socioeconomic inequality in adults undertaking HIV testing over time in Ethiopia based on data from demographic and health surveys

Aklilu Endalamaw[1,2]*, Charles F. Gilks[1], Yibeltal Assefa[1]

**1** School of Public Health, The University of Queensland, Brisbane, Australia, **2** College of Medicine and Health Sciences, Bahir Dar University, Bahir Dar, Ethiopia

* yaklilu12@gmail.com

**Data Availability Statement:** All relevant data are within the manuscript.

## Abstract

### Introduction

HIV testing is the entry point to HIV prevention, care and treatment and needs continuous evaluation to understand whether all social groups have accessed services equally. Addressing disparities in HIV testing between social groups results in effective and efficient response against HIV prevention. Despite these benefits, there was no previous study on inequality and determinants over time in Ethiopia. Thus, the objective of this research was to examine socioeconomic inequality in individuals undertaking HIV testing over time, allowing for the identification of persistent and emerging determinants.

### Methods

Data sources for the current study were the 2011 and 2016 Ethiopian Demographic Health Surveys. The 2016 population health survey is the one that Ethiopia used to set national AIDS response strategies; there was no other recent survey with HIV/AIDS-related indicators in Ethiopia. The final sample size for the current study was 28,478 for the year 2011 and 25,542 for the year 2016. The concentration curve and Erreygers' concentration index were used to estimate socioeconomic inequality in HIV testing. Subsequently, decomposition analysis was performed to identify persistent and emerging contributors of socioeconomic inequality. Generalized linear regression model with the logit link function was employed to estimate the marginal effect, elasticity, Erreygers' concentration index (ECI), and absolute and percentage contributions of each covariate.

### Results

The concentration curve was below the line of equality over time, revealing the pro-rich inequality in HIV testing. The inequality was observed in both 2011 (ECI = 0.200) and 2016 (ECI = 0.213). A household wealth rank had the highest percentage contribution (49.2%) for inequality in HIV testing in 2011, which increased to 61.1% in 2016. Additional markers include listening to the radio (13.4% in 2011 and 12.1% in 2016), education status (8.1% in

**Funding:** The author(s) received no specific funding for this work.

**Competing interests:** The authors have declared that no competing interests exist.

**Abbreviations:** AIDS, Acquired Immune Deficiency Syndrome; CC, Concentration Curve; CI, Confidence Interval; ECI, Erreygers' concentration index; EDHS, Ethiopia Demographic Health Survey; HIV, Human Immunodeficiency Virus; UNAIDS, United Nations Joint program for HIV/AIDS.

2011 and 6.8% in 2016), and resident (-2.0% in 2011 and 6.3% in 2016). Persistent determinants of individuals undertaking HIV testing were age 20–34 years, geographic region, education status, marital status, religion, income, media exposure (listening to the radio, reading newspaper, watching television), knowledge about HIV/AIDS, and attitudes towards people living with HIV. Age between 35 and 44 years and urban residence emerged as new associated factors in 2016.

## Conclusions

The higher HIV testing coverage was among individuals with higher socioeconomic status in Ethiopia. Socioeconomic inequality amongst individuals undertaking HIV testing was diverging over time. Household wealth rank, mass media exposure, education status, and resident took the largest share in explaining the disparity in individuals undertaking HIV testing between the lower and higher income groups. Therefore, interventions to equalise HIV testing coverage should take account of these determinants.

## Introduction

HIV testing serves as an entry point for providing subsequent health services for individuals infected with HIV [1]. It is one of the key strategies to the sustainable goal of ending the epidemic because a well- implemented HIV testing interventions can prevent a large number of HIV infections [2]. Many people acquire HIV infection from individuals who were unaware of their HIV status [3]. Screening individuals for HIV infection is one of the responses aimed at reducing new HIV infection to 250,000 by the end of 2030 globally [4].Universal HIV testing and treatment have the potential to improve life expectancy [5].

Therefore, the Joint United Nations Programme for HIV/AIDS (UNAIDS) envisions providing HIV testing for 95% of people living with HIV by the end of 2025 [6]. To achieve this, it has been emphasized that HIV testing services can be expanded with demand creation, counselling, self-testing, social networking, and retesting in high HIV-burden countries [7]. Moreover, community-based strategies, home-based interventions, and social-behavioural change communication were effective in increasing HIV testing [8]. Nations have also cascaded international strategies to their own socioeconomic and cultural contexts. For instance, Ethiopia has been implementing awareness creation, offering financial support for people living with HIV, providing behavioural change communication trainings, engaging the community members, and enhancing partnership [9, 10]. Ethiopia aspires to reduce the HIV incidence to lower than one per ten thousand population by the end of 2025 [11].

Despite inter-and intra-national interventions, global progress towards universal HIV testing has been 'off-track' and has varied between countries. According to the UNAIDS report, 3.3 million people were unreached for HIV testing in 2019 [12]. Additionally, as of 2021, 85% of people living with HIV knew their HIV status, leaving an additional 10% to be reached by 2025 [13]. In Ethiopia, only 9.2 million and 8.2 million HIV tests were conducted among the high-risk population in 2017 and 2018, respectively [14]. People's ability to pay for health care costs and other personal characteristics, such as education status, age, marital status, and residence, can influence the provision of HIV testing [14–16].

UNAIDS advises that conducting research is critical to addressing inequality in services and health problems [12]. Examining healthcare coverage and gaining a better understanding

of trends in socioeconomic inequality in HIV testing are important. Navigating healthcare coverage progress over time allows us to understand the added value or the persisted gaps due to the absence or presence of interventions [17]. Similarly, assessing trends in HIV testing coverage helps us understand who has access to HIV/AIDS services and who does not [18]. Previous studies have shown higher rates of HIV testing among wealthier groups [19–21], but not in all countries [22]. Addressing HIV testing coverage over time provides insights for allocating resources, considering policy implications, and understanding the persistent or emergent challenges [23, 24]. However, there have been no previous studies that have examined inequality in individuals undertaking HIV testing and its determinants over time in Ethiopia.

In Ethiopia, the HIV prevention roadmap was revised in 2018 with a due emphasis on reaching 90 percent of key and priority populations with combination HIV prevention, distributing two hundred millions condoms per year, and allocating one-fourth of the HIV/AIDS funding to HIV prevention [9]. The recent strategic plan includes a goal to achieve at least 95% people knowing their HIV status by 2025 through population campaigns, such as school-based campaigns, peer service providers, community outreach, and health facility HIV testing [14]. Understanding the inequality in HIV testing will, therefore, facilitate the implementation of the strategic plan.

Therefore, the aim of this study was to assess socioeconomic inequality and determinants in HIV testing over time in Ethiopia. The evidence from this study will provide the status of disparity over time, the main contributors of socioeconomic inequality in HIV testing, and identify the emergent and persistent determinants of HIV testing.

## Methods and materials

### Study design, setting, participants, and variables

A cross-sectional study was conducted based on the 2011 and 2016 population health survey in Ethiopia. The EDHS primarily collected health-related data among men aged 15 to 59 years and women aged 15 to 49 years using a multistage sampling technique. In 2011, the total sample size was 29,383, while it was 28,371 in 2016. For the current study, the participants considered were adults aged 15 to 49 years old. Since women aged 50 years and above were not included in the EDHS survey, we restricted the study population to individuals aged 15 to 49 years in this study. By excluding men aged 50 and above, the remaining participants in 2016 numbered 27,261. However, for the variable assessing accepting attitude towards people living with HIV (one of independent variables), individuals who had not heard about HIV/AIDS were excluded from answering accepting attitude towards people living with HIV-related questions. Thus, the final sample size became 25,542 for 2016 survey in this study. For the 2011 survey, missed data was handled by missing completely at random approach due to missed observations for some independent variables (employment status, reading newspaper, listening to the radio, and watching television). Additionally, like the 2016 survey year, men aged greater than or equal to 50 years and those not heard about HIV/AIDS were excluded. Then, the final sample sizes for the 2011 survey year became 28,478 in the current study.

The dependent variable was HIV testing, estimated as whether adults undergo HIV testing in the past 12 months preceding the survey date and received the results of the last test. Explanatory variables were age, sex, residence, geographic region, religion, employment status, education status, household wealth rank, sex of household head, comprehensive knowledge about HIV/AIDS, and attitude towards people living with HIV. To illustrate, age in years was categorized from 15 to 19, 20 to 24, 25 to 29, 30 to 34, 35 to 39, 40 to 44, and 45 to 49 including 15 and 49. Participants were grouped as men or women in sex while their sex of household head also coded as either men or women. Population distribution was grouped into rural or urban

based on residence while categorized into eleven groups geographically: Tigray, Afar, Amhara, Somali, Benshangul-Gumuz, Gambella, Oromia, Southern Nation Nationalities and People, Harari, Dire Dawa, and Addis Ababa. Study participants were asked about their religious affiliation and grouped to either Orthodox Christian, Muslim, Catholic, Protestant, Traditional and Others. Participants were asked whether they had been in employment for the last 12 months or not, and those who answered yes were understood as employed, otherwise no. The variables household wealth rank (richest, richer, moderate, poor, poorest) and education status (primary, secondary, tertiary, and higher) were categorical variables. Comprehensive knowledge about HIV/AIDS and accepting attitude towards people living with HIV was categorized as yes or no. The detail for comprehensive knowledge about HIV/AIDS [25] and accepting attitude towards people living with HIV is available elsewhere [26, 27].

To assure the quality of the data, EDHS undertook well-organised fieldwork activities, which involved, training and ongoing supervision using standardised and translated tools into national and local languages. The detail sampling technique, data quality assurance mechanism, measurement tools, and eligibility criteria are discussed in the 2011 and 2016 EDHS [26, 27]. After we obtained data from DHS website, proper data management, including appending women's and men's data, handling missed observation through missing completely at random, recoding, and variable recategorization, was properly conducted. The data were accessed from DHS on May 27, 2022, for research purposes.

## Statistical analysis

All analyses were performed using Stata software, version 17 (StataCorp LLC, College Station, TX, USA). The Erreygers' concentration index (ECI) was used to measure the socioeconomic inequality in HIV testing using '*conindex*' Stata command [28]. The ECI is intuitively shown through the concentration curve (CC). The CC plots the HIV testing coverage as a cumulative percentage (y-axis) against the cumulative percentage of the population ranked from poorest to richest (x-axis). The CC will be the line of equality (45-degree) when the same percentage of HIV testing occurs regardless of wealth index. The CC will lie below the line of equality when HIV testing is pro-rich or above the line of equality when HIV testing is pro-poor. The CC was drawn using '*glcurve*' stata command [29].

The CI is twice the area between the CC and the line of equality [30]. The value of CI is between −1 and 1; the exact value of −1 and 1 denotes absolute inequality, and a zero CI value represents equitable service distribution. A negative value indicates absolute inequality when the disproportionate concentration of HIV testing among the poor and a positive value of CI denotes the reverse. It is estimated from the covariance between HIV testing, and the fractional rank of the study participant by the wealth index. This is explained as:

$CI = \frac{2}{\mu}$ cov (yi, Ri), where CI is the concentration index for HIV testing; μ = the mean of HIV testing; yi = the dummy variable of whether the i participant tested for HIV; cov = the covariance with the sampling weights; Ri = the rank of the individual in the wealth distribution [31]. The standard CI may incorrectly estimate the extent of inequality when the health variable has a binary outcome [32]. The bounds of the standard CI for binary outcome are not -1 and 1. Because HIV testing has a binary outcome, the empirical bound of CI could be between m-1+ $(\frac{1}{n})$ and 1- m+ $(\frac{1}{n})$ [33]. In the analysis from large sample size, the value of $(\frac{1}{n})$ is close to zero. Therefore, the lower bound and the upper bounds of CI will be m-1 and 1- m, respectively [31]. To accommodate the bounded nature of the health variables with binary outcome, ECI is applied [34]. The ECI can be estimated as: ECI = 4m*CI, Where ECI is the Erreygers' corrected concentration index, CI is the generalised concentration index and m is the mean of the health variable, the proportion of HIV testing in this study. Erreygers' concentration index

was further decomposed into the contributions of each determinant to the income-related inequalities.

The contribution of each individual factor to the overall wealth-related inequality depends on two things. First, its impact on HIV testing (elasticity). Second, the degree of unequal distribution across different socioeconomic groups (CI). A factor that has a high impact but little variation across different wealth quintiles will contribute minimally to the overall inequality. Finally, we decomposed the ECI to obtain the contribution of covariates to the overall wealth-related inequality in HIV testing using methods suggested by Erreygers [33]. Because the decomposition of ECI depends on the assumption that health care is a linear function of the dependent variable, a suitable statistical analysis which allows categorical variables to be understood in a linear way is required. This understood categorical variable into a linear approach, which used a discrete change from 0 to 1 to use marginal or partial effects (dh/dx) estimates as:

$y = \Sigma_k \beta_k{}^m x_k + e$, where $\beta_k{}^m$ is the marginal effects (dh/dx) of each xk; e denotes the error term generated by the linear approximation. The regression-based decomposition was employed to examine the degree to which each covariate contributed to the inequality in HIV testing. The coefficients of contributors were included in the regression-based decomposition analysis. The generalised linear regression model (GLM) with a binomial distribution and a logit link function was analysed to get the regression coefficients of determinants. GLM decompose health variables with binary outcomes as a nonlinear regression model [35]. Therefore, based on above equation to determine absolute contributions, decomposition was conducted as:

$CI = \Sigma_k \left(\frac{\beta_{km}\bar{x}_k}{n}\right) CI_k + \left(\frac{GCe}{\mu}\right)$, where $\bar{x}_k$ stands for the mean of independent variables, $\beta_k{}^m$ represents the partial effect on independent variable xk (dy/dxk), $CI_k$ is the concentration index for determinant $x_k$, and GCIe is the generalised concentration index for the error term.

The regression model of the health dependent variables is performed for all $x_k$ to obtain the marginal effect of determinants, which is the association between the determinants and HIV testing (a positive sign indicates a positive association, while a negative sign indicates a negative association). Then, the elasticity of the health variables was calculated for each x ($x_k$), which is the sensitivity of HIV testing to changes in the determinants ($\frac{\beta_{km}\bar{x}_k}{n}$). It denotes the change in HIV testing associated with a one-unit change in the independent variables. Then, the CIs are estimated for HIV testing and each independent variable ($CI_k$). Finally, the contribution of each independent variable to the overall CI is estimated by multiplying the elasticity of each determinant by its CI, ($\frac{\beta_{km}\bar{x}_k}{n}$)$CI_k$.

## Results

### Participant characteristics of adults undergoing HIV testing

The percentage of HIV testing among adults aged 15 to 49 years was 20.8% in 2011 and 20.4% in 2016. With data in the two survey rounds (2011 and 2016), the aggregate-level percentage of adults who reported HIV testing was declined (0.4) percentage points difference). There was a large gap in reported HIV testing between urban and rural (19.7% absolute difference), non-educated and higher education (28.3% absolute difference), poorest and richest (25.4% absolute difference). The aggregate-level residential area-associated disparities in HIV testing increased between the two survey rounds, 17.4 percentage points in 2011 vs 19.7 percentage points in 2016. All age groups exhibited an increased disparity in HIV testing compared to 15 to 19 years old (e.g., 8.9 percentage points in 45–49 years old). Disparity was increased in employment status (2.7 percentage points) and richer group (3.5 percentage points) (Table 1).

**Table 1. Characteristics of participants and HIV testing among adults aged 15 to 49 years in Ethiopia between 2011 and 2016.**

| Variables | 2011 (n = 28,478) | | 2016 (n = 25,542) | |
|---|---|---|---|---|
| | Participants (%) | HIV testing (%) | Participants (%) | HIV testing (%) |
| Overall HIV test | | 20.8 | | 20.4 |
| **Age in years** | | | | |
| 15–19 | 6,750 (23.70) | 18.4 | 5,481 (21.46) | 11.6 |
| 20–24 | 5,109 (17.94) | 25.1 | 4,414 (17.28) | 24.9 |
| 25–29 | 5,290 (18.58) | 24.0 | 4,630 (18.13) | 27.0 |
| 30–34 | 3,441 (12.08) | 22.2 | 3,742 (14.65) | 23.0 |
| 35–39 | 3,481 (12.23) | 17.9 | 3,095 (12.12) | 19.4 |
| 40–44 | 2,311 (8.12) | 19.3 | 2,350 (9.20) | 19.8 |
| 45–49 | 2,096 (7.36) | 14.5 | 1,830 (7.16) | 15.9 |
| **Sex** | | | | |
| Male | 12,620 (44.31) | 20.9 | 11,160 (43.69) | 19.4 |
| Female | 15,858 (55.69) | 20.7 | 14,382 (56.31) | 21.1 |
| **Residence** | | | | |
| Urban | 6,768 (23.77) | 33.9 | 5624 (22.02) | 35.4 |
| Rural | 21,710 (76.23) | 16.8 | 19,918 (77.98) | 16.1 |
| **Region** | | | | |
| Tigray | 1,865 (6.55) | 33.08 | 1,789 (7.0) | 29.6 |
| Afar | 239 (0.84) | 18.9 | 194 (0.76) | 27.4 |
| Amhara | 7,649 (26.86) | 21.6 | 6,369 (24.93) | 22.5 |
| Oromia | 10, 544 (37.02) | 18.8 | 9,227 (36.12) | 16.3 |
| Somali | 495 (1.74) | 8.1 | 567 (2.22) | 10.8 |
| Benshangul-Gumuz | 295 (1.04) | 21.0 | 253 (0.99) | 25.4 |
| SNNPR | 5,480 (19.24) | 16.7 | 5,387 (21.09) | 16.9 |
| Gambela | 126 (0.44) | 28.8 | 73 (0.28) | 37.1 |
| Harari | 88 (0.31) | 29.6 | 63 (0.25) | 23.6 |
| Addis Ababa | 1,576 (5.53) | 32.6 | 1,474 (5.77) | 37.1 |
| Dire Dawa | 121 (0.43) | 41.6 | 146 (0.57) | 39.8 |
| **Education status** | | | | |
| No education | 11,567 (40.62) | 13.6 | 9,560 (37.43) | 14.6 |
| Primary | 12,859 (45.15) | 21.9 | 10,600 (41.50) | 18.4 |
| Secondary | 2,392 (8.40) | 35.6 | 3,531 (13.82) | 30.8 |
| Higher | 1,660 (5.83) | 41.3 | 1,851 (7.25) | 41.8 |
| **Marital status** | | | | |
| Never married | 9,800 (34.41) | 21.4 | 8,384 (32.83) | 16.3 |
| Widowed/divorced/no longer living together/separated | 16,624 (58.38) | 20.4 | 15,592 (61.05) | 22.0 |
| Married/living with partner | 2,054 (7.21) | 21.5 | 1,566 (6.13) | 25.7 |
| **Religion** | | | | |
| Orthodox | 13,653 (47.94) | 24.7 | 11,442 (44.80) | 25.1 |
| Catholic | 287 (1.01) | 19.1 | 184 (0.72) | 16.3 |
| Protestant | 5,966 (20.95) | 16.0 | 5,898 (23.09) | 15.9 |
| Muslim | 8,019 (28.16) | 18.6 | 7,686 (30.09) | 17.2 |
| Others and traditional | 552 (1.94) | 10.2 | 332 (1.30) | 13.3 |
| **Employment status** | | | | |
| Not employed | 7,273 (25.54) | 19.8 | 7,908 (30.96) | 17.9 |
| Employed | 21,205 (74.46) | 21.2 | 17,634 (69.04) | 21.5 |
| **Wealth Index** | | | | |

(*Continued*)

**Table 1.** (Continued)

| Variables | 2011 (n = 28,478) | | 2016 (n = 25,542) | |
|---|---|---|---|---|
| | Participants (%) | HIV testing (%) | Participants (%) | HIV testing (%) |
| Poorest | 4,842 (17.0) | 11.2 | 3933 (15.40) | 9.1 |
| Poorer | 5,212 (18.32) | 14.0 | 4,512 (17.66) | 12.5 |
| Middle | 5,276 (18.53) | 16.4 | 4,874 (19.08) | 15.9 |
| Richer | 5,767 (20.25) | 21.3 | 5313 (20.80) | 21.8 |
| Richest | 7,375 (25.90) | 34.8 | 6,911 (27.06) | 34.0 |
| **Reading newspaper** | | | | |
| No | 20,482 (71.92) | 16.7 | 20,309 (79.51) | 17.8 |
| Yes | 7,996 (28.08) | 31.3 | 5,233 (20.49) | 30.5 |
| **Listening to the radio** | | | | |
| No | 9,989 (35.08) | 14.0 | 14,719 (57.63) | 15.7 |
| Yes | 18,489 (64.92) | 24.5 | 10,823 (42.37) | 26.8 |
| **Watching television** | | | | |
| No | 13,580 (47.69) | 13.6 | 15,791 (61.82) | 15.2 |
| Yes | 14,898 (52.31) | 27.4 | 9,751 (38.18) | 28.8 |
| **Sex of household head** | | | | |
| Male | 22,959 (80.62) | 20.0 | 20,630 (80.77) | 19.8 |
| Female | 5,520 (19.38) | 24.4 | 4,912 (19.23) | 22.9 |
| **Comprehensive knowledge about HIV/AIDS** | | | | |
| No | 20,891 (73.36) | 17.6 | 18,049 (70.66) | 18.1 |
| Yes | 7,587 (26.64) | 29.9 | 7,493 (29.34) | 25.8 |
| **Accepting attitude towards people living with HIV** | | | | |
| No | 18,928 (66.47) | 16.0 | 15,389 (60.25) | 15.9 |
| Yes | 9,550 (33.53) | 30.5 | 10,153 (39.75) | 27.2 |

## Socioeconomic inequality in adults undergoing HIV testing

Fig 1 presents CC for HIV testing, and the corresponding ECIs are provided in Table 2. There was pro-rich inequality in HIV testing in Ethiopia in 2011 and 2016.

## Concentration index

The socioeconomic inequality in HIV testing increased from 0.200 in 2011 to 0.213 in 2016 (Table 2).

## Decomposition of ECI

The marginal effect showed that gender was a reciprocal determinant (men had better HIV testing access in 2011, while women had better access for HIV testing services in 2016). Education status (all categories than non-educated), household wealth rank (all categories than poorest), region (Tigray, Amhara, Gambela, Benshangul-Gumuz, and Dire Dawa have better than Addis Ababa), married rather than never married, Orthodox rather than protestant, mass media exposure (reading newspaper, listening to the radio, watching television), comprehensive knowledge about HIV/AIDS, and attitude towards people living with HIV were persistent determinants. The older age groups (20 to 24, 25 to 29, 30 to 34 years) had higher percentage point to HIV testing than 15 to19 years in both years.

A household wealth rank had the highest percentage contribution in inequality of adults undergoing HIV testing, 49.23% in 2011 which increased to 61.08% in 2016. The contribution

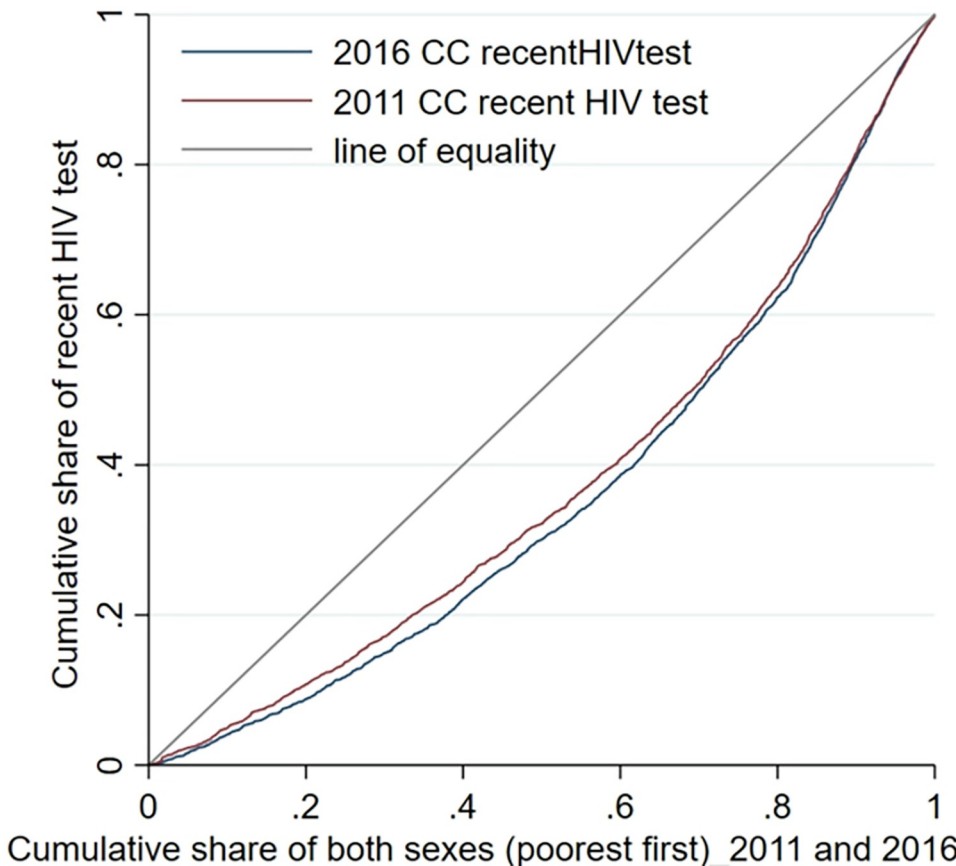

**Fig 1. Concentration curve (CC) of recent HIV test who received test result (HIV testing) among adults aged 15 to 49 years in Ethiopia in 2011 and 2016.**

of resident increased from -1.97% to 6.28%, while the contribution of education status, mass media exposure (reading newspaper, listening to radio, watching television), knowledge and attitude decreased over time despite its large contribution in each year (Table 3 and Fig 2).

## Discussion

This study examined inequality in adults undergoing HIV testing in 2011 and 2016 in Ethiopia using ECI and decomposition analysis. The finding showed that there was pro-rich inequality in those undertaking HIV testing that persisted over time.

The current study found that HIV testing rates were higher among the richest and higher-education populations compared to their counterparts. Our findings are consistent with those reported by Kim et al in Malawi [21], where the highest HIV test coverage was observed

**Table 2. Erreygers' concentration indices showing inequality in individuals undergoing HIV testing.**

| Outcome | | 2011 | 2016 |
|---|---|---|---|
| | | Adults 15 to 49 years | Adults 15 to 49 years |
| Recent HIV test | Erreygers' Concentration Index | 0.200*** | 0.213*** |
| | Sample size | 28,478 | 25,542 |

*** = P-value≤ 0.001

**Table 3. Marginal effects, elasticity, ECI, and contribution of variables among adults 15 to 49 years old in Ethiopia in 2011 and 2016.**

| Variables | 2011 | | | | | 2016 | | | | |
|---|---|---|---|---|---|---|---|---|---|---|
| | Marginal effects (95% CI) | Elasticity | Concentration Index (ECI) | Contribution | | Marginal effects (95% CI) | Elasticity | Concentration Index (ECI) | Contribution | |
| | | | | Value | Percentage | | | | Value | Percentage |
| **Age** (Ref:15–19) | | | | | | | | | | |
| 20–24 | 0.039 (0.024, 0.054)*** | 0.027 | 0.050 | 0.001 | 0.68 | 0.104(0.087, 0.121)*** | 0.071 | 0.031 | 0.002 | 1.04 |
| 25–29 | 0.043 (0.026, 0.060)*** | 0.031 | 0.011 | 0.0003 | 0.17 | 0.103 (0.084, 0.122)*** | 0.071 | 0.007 | 0.001 | 0.237 |
| 30–34 | 0.032 (0.012, 0.052)** | 0.015 | 0.004 | 0.0001 | 0.031 | 0.081 (0.060, 0.102)*** | 0.041 | -0.005 | -0.0002 | -0.105 |
| 35–39 | -0.004 (-0.025, 0.016) | -0.002 | -0.018 | 0.00004 | 0.018 | 0.050 (0.028, 0.072)*** | 0.023 | -0.018 | -0.0004 | -0.194 |
| 40–44 | 0.012 (-0.011, 0.035) | 0.004 | -0.019 | -0.0001 | -0.035 | 0.053 (0.030, 0.077)*** | 0.018 | -0.015 | -0.0003 | -0.129 |
| 45–49 | -0.021 (-0.046, 0.004) | -0.006 | -0.036 | 0.0002 | 0.11 | 0.016 (-0.010, 0.042) | 0.005 | -0.021 | -0.0001 | -0.046 |
| Summed | | | | 0.974 | | | | | 0.803 | |
| **Sex (Ref: Female)** | | | | | | | | | | |
| Male | 0.033 (0.022, 0.043)*** | 0.070 | -0.003 | -0.0002 | -0.10 | -0.027 (-0.038, -0.015)*** | -0.044 | -0.009 | 0.0004 | 0.196 |
| **Residence (Ref: rural)** | | | | | | | | | | |
| Urban | -0.007 (-0.025, 0.010) | -0.006 | 0.643 | -0.004 | -1.97 | 0.027 (0.009, 0.045)** | 0.022 | 0.601 | 0.013 | 6.28 |
| **Region** (Ref: Addis Ababa) | | | | | | | | | | |
| Tigray | 0.126 (0.103, 0.150)*** | 0.032 | -0.010 | -0.003 | -0.168 | 0.079 (0.055, 0.103)*** | 0.206 | -0.021 | -0.0004 | -0.205 |
| Afar | 0.038 (-0.017, 0.092) | 0.001 | -0.008 | -0.00001 | -0.01 | 0.105 (0.051, 0.158)*** | 0.004 | -0.013 | -0.00005 | -0.022 |
| Amhara | 0.070 (0.049, 0.090)*** | 0.076 | -0.114 | -0.009 | -4.34 | 0.023 (0.003, 0.044)* | 0.024 | -0.063 | -0.002 | -0.714 |
| Oromia | 0.020 (-0.0001, 0.040) | 0.029 | -0.019 | -0.001 | -0.28 | -0.029 (-0.050, -0.008)** | -0.043 | -0.031 | 0.001 | 0.626 |
| Somali | -0.109 (-0.162, -0.056)*** | -0.009 | -0.011 | 0.0001 | 0.048 | -0.042 (-0.087, 0.004) | -0.004 | -0.044 | 0.0002 | 0.089 |
| Benishangul-Gumuz | 0.063 (0.016, 0.111)** | 0.003 | -0.006 | -0.00002 | -0.01 | 0.062 (0.015, 0.110)* | 0.002 | -0.005 | -0.00001 | -0.006 |
| SNNPR | 0.024 (0.001, 0.047)* | 0.018 | -0.033 | -0.001 | -0.30 | 0.015 (-0.008, 0.038) | 0.012 | -0.027 | -0.0003 | -0.153 |
| Gambela | 0.086 (0.023, 0.150)** | 0.002 | 0.003 | 0.0000001 | 0.02 | 0.108 (0.032, 0.184)** | 0.002 | 0.0008 | 0.00001 | 0.001 |
| Harari | 0.021 (-0.052, 0.093) | 0.0002 | 0.007 | 0.000001 | 0.001 | -0.041 (-0.130, 0.048) | -0.0005 | 0.006 | -0.000003 | -0.001 |
| Dire Dawa | 0.112 (0.052, 0.171)*** | 0.002 | 0.009 | 0.00002 | 0.010 | 0.080 (0.026, 0.134)** | 0.002 | 0.010 | 0.00002 | 0.007 |

*(Continued)*

**Table 3.** (Continued)

| Variables | 2011 | | | | | 2016 | | | | |
|---|---|---|---|---|---|---|---|---|---|---|
| | Marginal effects (95% CI) | Elasticity | Concentration Index (ECI) | Contribution | | Marginal effects (95% CI) | Elasticity | Concentration Index (ECI) | Contribution | |
| | | | | Value | Percentage | | | | Value | Percentage |
| Summed | | | | -5.029 | | | | | -0.378 | |
| **Education** (Ref: no education) | | | | | | | | | | |
| Primary | 0.059 (0.047, 0.072)*** | 0.104 | 0.079 | 0.008 | 4.10 | 0.041 (0.028, 0.054)*** | 0.062 | 0.051 | 0.003 | 1.48 |
| Secondary | 0.088 (0.068, 0.108)*** | 0.028 | 0.172 | 0.005 | 2.40 | 0.087 (0.069, 0.104)*** | 0.038 | 0.224 | 0.008 | 3.99 |
| Higher | 0.087 (0.065, 0.109)*** | 0.019 | 0.166 | 0.003 | 1.62 | 0.088 (0.067, 0.110)*** | 0.018 | 0.151 | 0.003 | 1.28 |
| Summed | | | | | 8.12 | | | | | 6.75 |
| **Marital status (Ref: living together)** | | | | | | | | | | |
| Never married | -0.061 (-0.075, -0.047)*** | -0.080 | 0.173 | -0.014 | -6.9 | -0.100 (-0.115, -0.086)*** | -0.127 | 0.176 | -0.022 | -10.50 |
| Widowed/divorced/no longer living together/ separated | -0.022 (-0.041, -0.003)* | -0.006 | 0.007 | -0.00004 | -0.02 | 0.0001 (-0.020, 0.020) | 0.00001 | 0.006 | 0.0000001 | 0.00004 |
| Summed | | | | | -6.92 | | | | | -10.50 |
| **Religion (Ref: Orthodox)** | | | | | | | | | | |
| Catholic | -0.023 (-0.071, 0.024) | -0.001 | 0.001 | -0.000001 | -0.001 | -0.066 (-0.126, -0.007)* | -0.003 | 0.001 | -0.000004 | -0.002 |
| Protestant | -0.046 (-0.061, -0.031)** | -0.038 | -0.017 | 0.001 | 0.33 | -0.050 (-0.065, -0.034)*** | -0.041 | 0.014 | -0.001 | -0.280 |
| Muslim | 0.002 (-0.011, 0.014) | 0.002 | -0.086 | -0.0001 | -0.075 | -0.003 (-0.016, 0.010) | -0.004 | -0.160 | 0.001 | 0.276 |
| Others | -0.065 (-0.108, -0.022)** | -0.005 | -0.029 | 0.0001 | 0.075 | 0.005 (-0.045, 0.054) | 0.0004 | -0.029 | -0.00001 | -0.005 |
| Summed | | | | | 0.329 | | | | | -0.011 |
| **Occupation (Ref: Not employed)** | | | | | | | | | | |
| Employed | 0.004 (-0.008, 0.016) | 0.013 | -0.025 | -0.0003 | -0.162 | 0.009 (-0.003, 0.021) | 0.025 | -0.003 | -0.0001 | -0.030 |
| **Household wealth rank** (Ref: poorest) | | | | | | | | | | |
| Poorer | 0.024 (0.006, 0.042)** | 0.018 | -0.346 | -0.006 | -3.08 | 0.048 (0.027, 0.069)*** | 0.036 | -0.337 | -0.012 | -5.63 |
| Middle | 0.047 (0.029, 0.064)*** | 0.035 | -0.070 | -0.002 | -1.23 | 0.089 (0.069, 0.109)*** | 0.068 | -0.082 | -0.006 | -2.61 |
| Richer | 0.081 (0.064, 0.098)*** | 0.065 | 0.239 | 0.016 | 7.78 | 0.142 (0.122, 0.161)*** | 0.114 | 0.218 | 0.025 | 11.61 |
| Richest | 0.122 (0.100, 0.143)*** | 0.122 | 0.751 | 0.091 | 45.76 | 0.159 (0.137, 0.182)*** | 0.161 | 0.758 | 0.123 | 57.71 |
| Summed | | | | | 49.23 | | | | | 61.08 |

*(Continued)*

**Table 3.** (Continued)

| Variables | 2011 | | | | | 2016 | | | | |
|---|---|---|---|---|---|---|---|---|---|---|
| | Marginal effects (95% CI) | Elasticity | Concentration Index (ECI) | Contribution | | Marginal effects (95% CI) | Elasticity | Concentration Index (ECI) | Contribution | |
| | | | | Value | Percentage | | | | Value | Percentage |
| **Reading Newspaper (Ref: No)** | | | | | | | | | | |
| Yes | 0.031 (0.019, 0.042)*** | 0.033 | 0.362 | 0.012 | 6.04 | 0.023 (0.010, 0.036)*** | 0.021 | 0.324 | 0.007 | 3.20 |
| **Listening to radio (Ref: No)** | | | | | | | | | | |
| Yes | 0.033 (0.022, 0.045)*** | 0.085 | 0.313 | 0.027 | 13.37 | 0.037 (0.026, 0.048)*** | 0.076 | 0.339 | 0.026 | 12.13 |
| **Watching television (Ref: No)** | | | | | | | | | | |
| Yes | 0.057 (0.046, 0.068)*** | 0.116 | 0.12 | 0.048 | 24.00 | 0.015 (0.003, 0.028)* | 0.024 | 0.433 | 0.010 | 4.88 |
| **Sex of household head (Ref: Male)** | | | | | | | | | | |
| Female | 0.018 (0.006, 0.030)** | 0.013 | 0.086 | 0.001 | 0.57 | 0.007 (-0.006, 0.020) | 0.005 | 0.084 | 0.0004 | 0.209 |
| **Comprehensive knowledge of HIV/AIDS (Ref: No)** | | | | | | | | | | |
| Yes | 0.040 (0.030, 0.050)*** | 0.042 | 0.237 | 0.010 | 4.94 | 0.019 (0.008, 0.0295)*** | 0.019 | 0.230 | 0.004 | 2.05 |
| **Attitude towards people living with HIV** (Ref: No) | | | | | | | | | | |
| Yes | 0.032 (0.021, 0.043)*** | 0.042 | 0.432 | 0.018 | 9.01 | 0.0189 (0.008, 0.0297)*** | 0.010 | 0.177 | 0.002 | 0.854 |

Ref = reference

among individuals with a higher wealth index. Household wealth rank was identified as the most significant contributing factor for this difference. In Africa, health care services have primarily been used by the wealthier population [36]. It has been pointed out in the current and previous studies that demographic and behavioural factors can contribute to lower rates of HIV testing among individuals from lower-income households [37]. The inability to afford the cost of HIV testing may be a reason why individuals in lower income households do not undergo HIV testing at the same rate as wealthier individuals. In Ethiopia, the costs per patient for voluntary counselling and testing were US$5.06 for home-based testing, US$6.55 for voluntary counselling and testing, and US$3.35 for professionally initiated counselling and testing [38]. This implies that individuals in lower socioeconomic households might face higher costs relative to their monthly income or be unable to afford the estimated HIV-testing costs, compared to richest individuals who have better financial resources. Therefore, it is necessary to expand HIV testing services targeted at lower-income groups. Strategies, such as increasing health care insurance may assist the poorest to cover the HIV testing expenditure. Nonetheless, health insurance coverage is low in sub-Saharan African countries except in individuals with a higher wealth status [39]. Similarly, health insurance coverage is low in Ethiopia [40].

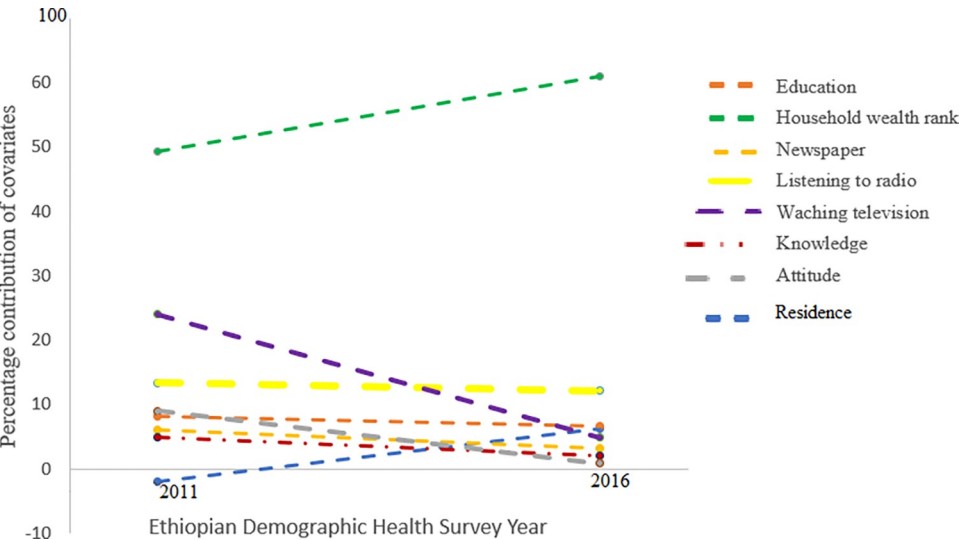

**Fig 2. Percentage contribution of covariates to socioeconomic inequality in HIV testing in Ethiopia between 2011 and 2016.**

It is important to note that education status significantly contributed to socioeconomic inequality in HIV testing in Ethiopia. This finding aligns with a study in Vietnam [41] and another in Malawi [21], demonstrating consistency across different contexts. As individuals' education levels increase, they experience benefits that extend beyond mere changes in knowledge and behaviour. For example, higher education attainment is often accompanied by an increase in wealth status [42]. Moreover, educated individuals tend to possess a greater capacity to make informed decisions regarding health services [43], resulting in higher rates of HIV testing among those who are more educated and financially well-off. As a result, the presence of difference in wealth status between uneducated and educated triggered socioeconomic inequality in HIV testing. This issue requires careful policy attention and strategic interventions. The Ethiopian 2021 to 2025 National Health Policy Strategy Plan includes a school health program as a tool for comprehensive HIV prevention. However, it does not mention a special plan for uneducated (school unattended) and lower educated individuals outside of the school environment [14]. In health care settings, for example, health professionals may adopt a unique approach to deliver HIV testing for uneducated clients if a well-established system is in place.

Our result suggested that the contribution of media exposure in HIV testing is notable and persisted over time. Despite an increasing mobile-technology with internet access, printed and electronic mass-media are continuing important social-behavioural change determinants. The Ethiopian health policy gives emphasis on HIV education to increase HIV testing coverage through mass media, social media, and interactive digital applications targeting key and priority populations such as girls and young women, female sex workers, workers in hotspot areas. However, it does not give emphasise to the poor [14]. It would be helpful if the usage of mass-media for health programmes could also accommodate individuals with lower socioeconomic status.

Knowledge about HIV/AIDS and attitude towards people living with HIV also contributed importantly. The current study also affirms the importance of taking social and behavioural change education in health policy because a high or low HIV testing usage is attributed by comprehensive knowledge about HIV/AIDS and attitude towards people living with HIV.

This might be due to the fact that stereotyping people living with HIV is higher among people who live in low socioeconomic status, which limits HIV testing frequency and coverage [44].

Generally, multi-locational and sustainable initiatives are important, including social behaviour change, such as health care providers interaction with people living with HIV, social inclusion, reduction of stereotyping others, emotionally engaging content in media production and employment for unemployed adults. Ethiopian set out a drafted national strategy from 2021 to 2025 to increase HIV testing coverage using 'key and priority population through friendly clinics', 'Drop-in Centres', 'integrated harm reduction services', 'HIV services for prisoners', 'integrated HIV services at hot spot workplaces', 'peer service providers program', and 'HIV mainstreaming'[14]. These service areas should prioritize the recognition and consideration of socioeconomic inequality and its major contributors. In this strategy document [14], emphasis is placed on high HIV risk area, high-risk populations, and gender inequality have got emphasis, while inequalities related to socioeconomic status, education, religion, and employment classes are not taken into consideration.

## Limitation

This population-based study was based on 2011 and 2016 survey. Ethiopia has not conducted a population-based survey on HIV/AIDS indicators since 2016, and the country uses data from the 2016 survey for policy initiation and program implementation. Additionally, this decomposition analysis cannot establish a cause-effect relationship between exploratory and outcome variables; it primarily shows the correlation between socioeconomic and HIV testing was measured based on self-reports from the participants, which may be subject to bias.

## Conclusions

The higher percentage of HIV testing was among individuals with a higher socioeconomic status. This socioeconomic inequality had been diverged over time. Household wealth rank, mass media exposure, education status, and residence contribute to the significant socioeconomic inequality over time. It is important to implement an individual and public-health approach embedded with the primary health care principles that are addressed through community engagement and multisectoral collaboration. Therefore, pro-poor policy initiatives should emphasise a tailored and need-based sustainable approach to addressing social determinants.

## Acknowledgments

We would like to thank the Demographic Health Survey for giving the dataset to conduct this research.

## Author Contributions

**Conceptualization:** Aklilu Endalamaw.

**Data curation:** Aklilu Endalamaw.

**Formal analysis:** Aklilu Endalamaw.

**Investigation:** Aklilu Endalamaw.

**Methodology:** Aklilu Endalamaw.

**Project administration:** Aklilu Endalamaw.

**Software:** Aklilu Endalamaw.

**Supervision:** Aklilu Endalamaw, Charles F. Gilks, Yibeltal Assefa.

**Validation:** Aklilu Endalamaw, Charles F. Gilks, Yibeltal Assefa.

**Visualization:** Aklilu Endalamaw, Yibeltal Assefa.

**Writing – original draft:** Aklilu Endalamaw.

**Writing – review & editing:** Aklilu Endalamaw, Charles F. Gilks, Yibeltal Assefa.

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
