## [Decision Letter · Decision Letter 0]

10 Nov 2023

PONE-D-23-30633Socioeconomic inequality in HIV testing over time in Ethiopia: a population-based studyPLOS ONE

Dear Dr. Endalamaw,

Thank you for submitting your manuscript to PLOS ONE. After careful consideration, we feel that it has merit but does not fully meet PLOS ONE’s publication criteria as it currently stands. Therefore, we invite you to submit a revised version of the manuscript that addresses the points raised during the review process.

Clearly state the sampling size step by step. Check for the grammar and sentence structure==============================

Please submit your revised manuscript by Dec 25 2023 11:59PM. If you need more time than this to complete your revisions, please reply to this message or contact the journal office at plosone@plos.org. Please include the following items when submitting your revised manuscript:A rebuttal letter that responds to each point raised by the academic editor and reviewer(s). You should upload this letter as a separate file labeled 'Response to Reviewers'.A marked-up copy of your manuscript that highlights changes made to the original version. You should upload this as a separate file labeled 'Revised Manuscript with Track Changes'.An unmarked version of your revised paper without tracked changes. You should upload this as a separate file labeled 'Manuscript'.If applicable, we recommend that you deposit your laboratory protocols in protocols.io to enhance the reproducibility of your results. Protocols.io assigns your protocol its own identifier (DOI) so that it can be cited independently in the future. For instructions see: https://journals.plos.org/plosone/s/submission-guidelines#loc-laboratory-protocols. Additionally, PLOS ONE offers an option for publishing peer-reviewed Lab Protocol articles, which describe protocols hosted on protocols.io. Read more information on sharing protocols at https://plos.org/protocols?utm_medium=editorial-email&utm_source=authorletters&utm_campaign=protocols.

We look forward to receiving your revised manuscript.

Kind regards,

Kendalem Asmare Atalell, MSc

Academic Editor

PLOS ONE

Journal Requirements:

3. Thank you for submitting the above manuscript to PLOS ONE. During our internal evaluation of the manuscript, we found significant text overlap between your submission and previous work in the [introduction, conclusion, etc.].

Please revise the manuscript to rephrase the duplicated text, cite your sources, and provide details as to how the current manuscript advances on previous work. Please note that further consideration is dependent on the submission of a manuscript that addresses these concerns about the overlap in text with published work.

[If the overlap is with the authors’ own works: Moreover, upon submission, authors must confirm that the manuscript, or any related manuscript, is not currently under consideration or accepted elsewhere. If related work has been submitted to PLOS ONE or elsewhere, authors must include a copy with the submitted article. Reviewers will be asked to comment on the overlap between related submissions (http://journals.plos.org/plosone/s/submission-guidelines#loc-related-manuscripts).]

We will carefully review your manuscript upon resubmission and further consideration of the manuscript is dependent on the text overlap being addressed in full. Please ensure that your revision is thorough as failure to address the concerns to our satisfaction may result in your submission not being considered further.

Additional Editor Comments:

Dear Aklilu, thank you for submitting your work in PLOS one. Your work is great and would be a good addition to the scientific world towards the association between socioeconomic inequalities and trends in HIV testing.

Before proceeding to the next level you have to address the issues raised by the academic reviewers.

Kind regards

Kendalem

Academic editor

Reviewers' comments:

Reviewer's Responses to Questions

**Comments to the Author**

1. Is the manuscript technically sound, and do the data support the conclusions?

Reviewer #1: Partly

Reviewer #2: Yes

2. Has the statistical analysis been performed appropriately and rigorously? 

Reviewer #1: Yes

Reviewer #2: Yes

3. Have the authors made all data underlying the findings in their manuscript fully available?

Reviewer #1: Yes

Reviewer #2: Yes

4. Is the manuscript presented in an intelligible fashion and written in standard English?

Reviewer #1: No

Reviewer #2: Yes

5. Review Comments to the Author

Reviewer #1: 1. The introduction is generally fine but needs to be improved. It is very important that the grammatical construction of the section is improved to make it more easily readable.

2. While the authors suggest that their study contributes to the literature, this is not properly demonstrated. What did previous studies miss out that your study contributes to? Why is a trend analysis important in this context? It is not enough to state these claims, they should be demonstrated.

3. It is also important to demonstrate why one should expect inequality in testing. What is current practice with regards to testing? Are there outreach campaigns? Who is targeted? What are national policies with regards to testing? In sum more context is needed.

4. I am sure the lack of clear understanding in the sample description may be due to the poor grammar but there are several questions that need to be answered for clarity. I have listed some of these below.

• What motivated the 15-49 age bracket? Is it just the data or some context specific conditions?

• You refer to “final sample size” how did you get to this final? Where did you start from?

• Because the make up of the sample is not clear I also wonder who is included. The objective of the paper requires that to correctly understand the nature of inequality the right people should be included in the sample. For instance, HIV testing is typically not common practice. People test only when they feel exposed or at risk. I therefore wonder if this sample includes people at risk or just the general population.

5. In the methods, a section should be dedicated to defining the variables used in the analysis. This is missing.

6. It was also mentioned that the ECI was decomposed. I wonder how this was done. Maybe you could mention which Stata command was used for this decomposition.

7. Reading through the paper, it does seem like using two different surveys has no additional benefit to the study. If there is no relevance, then I suggest you only stick with the latest survey year. It should be noted that for a computational analysis of this nature, changes over time cannot be interpreted as increase or decrease especially because the sample differ for the different survey years.

8. The discussion should be improved and aligned with current policies in the country.

9. The limitations should include the fact that the measure of testing is self-reported and may be subject to bias.

Reviewer #2: Thank you for submitting your article. Overall, your work is promising and contributes significantly to the understanding of this important issue.

I appreciate the clarity of your title and the organization of your manuscript. However, I would recommend addressing the following:

While your literature review of cross-sectional surveys in sub-Saharan African countries is informative, consider comparing your findings with smaller studies in other contexts, such as Hajizadeh et al. (2014), to provide a more comprehensive perspective. (Hajizadeh, M., Sia, D., Heymann, S.J. et al. Socioeconomic inequalities in HIV/AIDS prevalence in sub-Saharan African countries: evidence from the Demographic Health Surveys. Int J Equity Health 13, 18 (2014). https://doi.org/10.1186/1475-9276-13-18)

In the 'Conclusion' section, try to succinctly summarize the practical implications of your research for policymakers and public health practitioners.

Make sure to proofread your manuscript for grammar and clarity.

In addition, I believe it would be beneficial for readers if you added to the title that the data originates from the Ethiopian Demographic and Health Surveys (EDHS). This would help emphasize that you've followed an internationally validated DHS methodology and provide a better understanding of the data quality used in your study.

6. PLOS authors have the option to publish the peer review history of their article (what does this mean?). If published, this will include your full peer review and any attached files.

Reviewer #1: No

Reviewer #2: **Yes: **Marilina Santero

---

## [Author Response · Author response to Decision Letter 0]

27 Nov 2023

Manuscript submission ID: PONE-D-23-30633

Socioeconomic inequality in adults undertaking HIV testing over time in Ethiopia based on Demographic Health Surveys

Dear Plos One Academic Editor,

Thank you for providing us the opportunity to revise our manuscript based on helpful comments and suggestions. We appreciate the time and effort you, and reviewers dedicate to providing comments and suggestions. In our revision submission, we have added supplement ideas, made corrections, and explained some notes based on the academic editor’s and two reviewers’ comments. We have highlighted the changes in red colour in the manuscript submitted with track change.

The following are the authors’ point-by-point responses (the reviewers’ comments are numbered, and the authors’ responses are underlined in italic format). 

Reviewer #1

1. The introduction is generally fine but needs to be improved. It is very important that the grammatical construction of the section is improved to make it more easily readable.

Authors’ response: Thank you for your suggestions. We have extensively edited the revised submission. 

2. While the authors suggest that their study contributes to the literature, this is not properly demonstrated. What did previous studies miss out that your study contributes to? Why is a trend analysis important in this context? It is not enough to state these claims, they should be demonstrated.

Authors’ response: Thank you for your insights. We have now reasoned the importance of conducting trend analysis. Examining healthcare coverage and gaining a better understanding of trends in socioeconomic inequality in HIV testing are important. Navigating healthcare coverage progress over time allows us to understand the added value or the persisted gaps due to the absence or presence of interventions (17). Similarly, assessing trends in HIV testing coverage helps us understand who has access to HIV/AIDS services and who does not (18). Previous studies have shown higher rates of HIV testing among wealthier groups (19-21), although contradictory findings have been reported in South Africa (22). Addressing HIV testing coverage over time provides insights for allocating resources, considering policy implications, and understanding the persistent or emergent challenges (23, 24). However, there have been no previous studies examined inequality in HIV testing and its determinants over time in Ethiopia.

3. It is also important to demonstrate why one should expect inequality in testing. What is current practice with regards to testing? Are there outreach campaigns? Who is targeted? What are national policies with regards to testing? In sum more context is needed.

Authors’ response: Thank you for raising these issues. We have added more contexts in the revised submission. In Ethiopia, the HIV prevention roadmap was revised in 2018 with a due emphasis on reaching 90 percent of key and priority populations with combination HIV prevention, distributing two hundred millions condoms per year, and allocating one-fourth of the HIV/AIDS funding to HIV prevention (25). The recent strategic plan includes a goal to achieve at least 95% people knowing their HIV status by 2025 through population campaigns, such as school-based campaigns, peer service providers, community outreach, health facility HIV testing (14). Understanding the inequality in HIV testing will, therefore, facilitate the implementation of the strategic plan. 

4. I am sure the lack of clear understanding in the sample description may be due to the poor grammar but there are several questions that need to be answered for clarity. I have listed some of these below.

• What motivated the 15-49 age bracket? Is it just the data or some context specific conditions?

• You refer to “final sample size” how did you get to this final? Where did you start from?

• Because the make up of the sample is not clear I also wonder who is included. The objective of the paper requires that to correctly understand the nature of inequality the right people should be included in the sample. For instance, HIV testing is typically not common practice. People test only when they feel exposed or at risk. I therefore wonder if this sample includes people at risk or just the general population.

Authors’ response: Thank you for asking clarification. We have described why the study population limited to 15 to 49 years and how the final sample size estimated in the revised submission. The EDHS primarily collected health-related data among men aged 15 to 59 years and women aged 15 to 49 years using a multistage sampling technique. In 2011, the total sample size was 29,383, while it was 28,371 in 2016. For the current study, the participants considered were adults aged 15 to 49 years old. Since women aged 50 years and above were not included in the EDHS survey, we restricted the study population to individuals aged 15 to 4y years in this study. By excluding men aged 50 and above, the remaining participants in 2016 numbered 27,261. However, for the variable assessing accepting attitude towards people living with HIV (one of independent variables), individuals who were not heard about HIV/AIDS were excluded from answering accepting attitude towards people living with HIV-related questions. Thus, the final sample size became 25,542 for 2016 survey in this study. For the 2011 survey, missed data was handled by missing completely at random approach due to missed observations for some independent variables (employment status, reading newspaper, listening to the radio, and watching television). Additionally, like the 2016 survey year, men aged greater than or equal to 50 years and those not heard about HIV/AIDS were excluded. Then, the final sample sizes for the 2011 survey year became 28,478 in the current study.

5. In the methods, a section should be dedicated to defining the variables used in the analysis. This is missing.

Authors’ response: Thank you for reminding us. We have described the detail of each included variables in the method section in the revised submission. The dependent variable was HIV testing, estimated as whether adults undergo HIV testing in the past 12 months preceding the survey date and received the results of the last test. Explanatory variables were age, sex, residence, geographic region, religion, employment status, education status, household wealth rank, sex of household head, comprehensive knowledge about HIV/AIDS, and attitude towards people living with HIV. To illustrate, age in years was categorized from 15 to 19, 20 to 24, 25 to 29, 30 to 34, 35 to 39, 40 to 44, and 45 to 49 including 15 and 49. Participants were grouped as men or women in sex while their sex of household head also coded as either men or women. Population distribution was grouped into rural or urban based on residence while categorized into eleven groups geographically: Tigray, Afar, Amhara, Somali, Benshangul-Gumuz, Gambella, Oromia, Southern Nation Nationalities and people, Harari, Dire Dawa, and Addis Ababa. Study participants were asked about their religious affiliation and grouped to either Orthodox Christian, Muslim, Catholic, Protestant, and Traditional and Others. Participants were asked whether they had employed for the last 12 months or not, and those who answered yes were understood as employed, otherwise no. The variables household wealth rank (richest, richer, moderate, poor, poorest) and education status (primary, secondary, tertiary, and higher) were categorical variables. Comprehensive knowledge about HIV/AIDS and accepting attitude towards people living with HIV was categorized as yes or no. The detail for comprehensive knowledge about HIV/AIDS (26) and accepting attitude towards people living with HIV is available elsewhere (27, 28).

6. It was also mentioned that the ECI was decomposed. I wonder how this was done. Maybe you could mention which Stata command was used for this decomposition.

Authors response: The Erreygers’ concentration index (ECI) was used to measure the socioeconomic inequality in HIV testing using ‘conindex’ Stata command (29). The CC was drawn using ‘glcurve’ stata command (30).

7. Reading through the paper, it does seem like using two different surveys has no additional benefit to the study. If there is no relevance, then I suggest you only stick with the latest survey year. It should be noted that for a computational analysis of this nature, changes over time cannot be interpreted as increase or decrease especially because the sample differ for the different survey years.

Authors’ response: Examining healthcare coverage and gaining a better understanding of trends in socioeconomic inequality in HIV testing are relevant. Navigating healthcare coverage progress over time allows us to understand the added value or the persisted gaps due to the absence or presence of interventions. Similarly, assessing trends in HIV testing coverage helps us understand who has access to HIV/AIDS services and who does not. Addressing HIV testing coverage over time provides insights for allocating resources, considering policy implications, and understanding the persistent or emergent challenges. 

 We agree that the sample survey years are different. However, we highlighted the importance of addressing trend because both surveys are population based and representative of the population in the same country. Moreover, Ethiopia has utilized the different Demographic Health Survey years’ report to monitor the progresses in different health indicators.

8. The discussion should be improved and aligned with current policies in the country.

Authors’ response: Thank you so much. We have made some adjustment and we have considered the current policies of the country. 

9. The limitations should include the fact that the measure of testing is self-reported and may be subject to bias.

Authors’ response: Thank you for the suggestion. We have added that HIV testing was measured based on self-reports from the participants, which may be subject to bias.

Reviewer #2: 

1. Thank you for submitting your article. Overall, your work is promising and contributes significantly to the understanding of this important issue.

Authors’ response: Thank you for appreciating our work. 

2. I appreciate the clarity of your title and the organization of your manuscript. However, I would recommend addressing the following:

Authors’ response: Thank you for appreciation and suggestion. We have revised the topic as Socioeconomic inequality in HIV testing over time in Ethiopia based on Demographic Health Surveys 

3. While your literature review of cross-sectional surveys in sub-Saharan African countries is informative, consider comparing your findings with smaller studies in other contexts, such as Hajizadeh et al. (2014), to provide a more comprehensive perspective. (Hajizadeh, M., Sia, D., Heymann, S.J. et al. Socioeconomic inequalities in HIV/AIDS prevalence in sub-Saharan African countries: evidence from the Demographic Health Surveys. Int J Equity Health 13, 18 (2014). https://doi.org/10.1186/1475-9276-13-18)

Authors’ response: Thank you so much for this suggestion. We have tried to discuss the available articles. We really acknowledge for the articles (‘socioeconomic inequality in HIV prevalence’) you suggested to include in our discussion. We may not consider this article in the current manuscript because our current study is focused on socioeconomic inequality in HIV testing. However, I assure you that we will consider discussing socioeconomic inequality in HIV prevalence by comparing with the Ethiopian context in another study because we have also conducted another study on socioeconomic inequality in HIV prevalence. 

4. In the 'Conclusion' section, try to succinctly summarize the practical implications of your research for policymakers and public health practitioners.

Authors’ response: Thank you for the suggestion. We have considered implications in the revised submission. 

5. Make sure to proofread your manuscript for grammar and clarity.

Authors’ response: Thank you so much for this suggestion. We have extensively edited the revised submission. 

6. In addition, I believe it would be beneficial for readers if you added to the title that the data originates from the Ethiopian Demographic and Health Surveys (EDHS). This would help emphasize that you've followed an internationally validated DHS methodology and provide a better understanding of the data quality used in your study.

Authors’ response: We have revised the title as Socioeconomic inequality in HIV testing over time in Ethiopia based on Demographic Health Surveys

---

## [Editor Report · Decision Letter 1]

20 Dec 2023

Socioeconomic inequality in adults undertaking HIV testing over time in Ethiopia based on data from Demographic and Health Surveys

PONE-D-23-30633R1

Dear Dr. Aklilu

We’re pleased to inform you that your manuscript has been judged scientifically suitable for publication and will be formally accepted for publication once it meets all outstanding technical requirements.

Kind regards,

Kendalem Asmare Atalell, MSc

Academic Editor

PLOS ONE
---

## [Editor Report · Acceptance letter]

6 Feb 2024

PONE-D-23-30633R1 

PLOS ONE

Dear Dr. Endalamaw, 

I'm pleased to inform you that your manuscript has been deemed suitable for publication in PLOS ONE. Congratulations! Your manuscript is now being handed over to our production team.

Kind regards, 

on behalf of

Dr. Kendalem Asmare Atalell 

Academic Editor

PLOS ONE